# Differential Expression of Genes Related to Growth and Aflatoxin Synthesis in *Aspergillus flavus* When Inhibited by *Bacillus velezensis* Strain B2

**DOI:** 10.3390/foods11223620

**Published:** 2022-11-13

**Authors:** Qiaoyun Wu, Huanhuan Li, Sunxing Wang, Zhongnian Zhang, Zhipeng Zhang, Tuwei Jin, Xiufang Hu, Guohong Zeng

**Affiliations:** Zhejiang Province Key Laboratory of Plant Secondary Metabolism and Regulation, College of Life Science and Medicine, Zhejiang Sci-Tech University, Road 2, Xiasha, Hangzhou 310018, China

**Keywords:** *Aspergillus flavus*, *Bacillus velezensis* strain B2, antagonistic activity, transcriptome analysis, aflatoxin

## Abstract

*Aspergillus flavus* is a saprophytic soil fungus that infects and contaminates seed crops with the highly carcinogenic aflatoxin, which brings health hazards to animals and humans. In this study, bacterial strains B1 and B2 isolated from the rhizosphere soil of *camellia sinensis* had significant antagonistic activities against *A. flavus.* Based on the phylogenetic analysis of 16SrDNA gene sequence, bacterial strains B1 and B2 were identified as *Bacillus tequilensis* and *Bacillus velezensis,* respectively. In addition, the transcriptome analysis showed that some genes related to *A. flavus* growth and aflatoxin synthesis were differential expressed and 16 genes in the aflatoxin synthesis gene cluster showed down-regulation trends when inhibited by *Bacillus velezensis* strain B2. We guessed that the *Bacillus velezensis* strain B2 may secrete some secondary metabolites, which regulate the related gene transcription of *A. flavus* to inhibit growth and aflatoxin production. In summary, this work provided the foundation for the more effective biocontrol of *A. flavus* infection and aflatoxin contamination by the determination of differential expression of genes related to growth and aflatoxin synthesis in *A. flavus* when inhibited by *B. velezensis* strain B2.

## 1. Introduction

*Aspergillus flavus* is a saprophytic soil fungus that infects and contaminates seed crops and it is in the spotlight for the production of aflatoxins that contaminate oil-rich seeds such as maize, peanuts, cotton seeds and treenuts before and after harvest. *A. flavus*, as the second *Aspergillus fumigatus* only pathogen, produces secondary metabolites containing polyketones, which are strong carcinogens and can seriously affect human and animal health [1]. Aflatoxins are one of the most harmful mycotoxins, which are produced by *A. flavus*, *A. parasiticus*, and other fungi that are commonly found in the production and preservation of grain and feed. Aflatoxins can cause harm to animal and human health due to their toxic (carcinogenic, teratogenic, and mutagenic) effects [2]. Aflatoxins were first discovered and characterized in the early 1960s following a severe livestock poisoning incident in England involving turkeys; they are kinds of polyketide-derived carcinogenic and mutagenic secondary metabolites [3,4,5]. Aflatoxins belong to a class of secondary metabolites of *A. flavus* and include varieties and the four major types are B1, B2, G1 and G2. Aflatoxin B1 (AFB1) exerts strong mutagenicity and carcinogenicity [6], which is classified as a Group 1 human carcinogen by the International Agency for Research on Cancer [7,8]. It is converted to AFB1-8 and 9-epoxide in the liver, catalyzed by cytochrome P450 associated enzymes after being ingested by animals, forming adducts with the guanine base of DNA, thus resulting in acute and chronic diseases called aflatoxicoses in both human and household animals feeding high levels of aflatoxins-contaminated food [9]. AFB1 biosynthesis was catalyzed by the coordinated cascade of enzymes that are encoded by approximately 30 different genes that group into a cluster located near the telomeric region of chromosome three of the aflatoxinogenic species [10]. The initial stage of aflatoxin biosynthesis is similar to fatty acid biosynthesis, with acetyl-CoA as the starting unit and malonyl-CoA as the elongating unit, which is catalyzed by polyketide synthase (PKSA) to form the aflatoxin of the polyketone backbone [11].

Aflatoxin not only harms the health of human and animals, but also causes economic losses. Human and animals themselves cannot degrade aflatoxin through their own metabolism and so avoid contact with or ingestion of aflatoxin [12]. The pollution of aflatoxin to grain occurs in all stages before harvest, during harvest, storage and processing. Therefore, at each stage, from planting to harvest, there are different methods for the prevention and control of aflatoxin. The prevention and control of aflatoxin can be divided into prenatal and postpartum prevention and control [13]. At present, the main method to control the pollution of *A. flavus* before harvest is to use the non-toxigenic-strain of *A. flavus* to compete with the toxigenic-strain of *A. flavus* for the nutrient substrate to achieve inhibition [14,15]. COTTYPJ et al. [16] applied the non-toxigenic-strain of *A. flavus* to cotton fields to effectively reduce aflatoxin pollution in cotton seeds. The effect of the field biological control of non-toxin-producing *A. flavus* strains can last until harvest and storage [17]. At present, large-scale research has led to the rapid development of non-toxigenic-strains of *A. flavus* in commercial application [18,19]. The control measures of *A. flavus* and its toxin in post-harvest crops mainly include physical control, chemical control and biological control [20]. Physical methods mainly include ozone, irradiation, high temperature heating and so on, while chemical methods include ammoniation, alkali, oxidation and so on. However, these two methods have some shortcomings, such as an unstable effect, large loss of nutrients and difficulty in large-scale production [21]. The biological control of *A. flavus* is a method to control *A. flavus* and its toxin using microorganisms and their secondary metabolites or enzymes is secreted. A large number of studies have shown that many microorganisms can inhibit the growth and toxin production of *A. flavus*, including bacteria, actinomycetes, yeast, algae and so on [22]. During their growth, they usually produce a series of secondary metabolites, including alcohols, aldehydes, acids, esters and ketones [23,24]. At present, scholars at home and abroad mainly study the volatile substances produced by *Bacillus*, *Pseudomonas* and *Streptomyces*. The volatile compounds produced by plant rhizosphere bacteria can not only inhibit the growth and toxin production of *A. flavus*, but also have a certain effect on plant growth [25]. For example, volatile substances produced by *Bacillus amylolyticus* can effectively stimulate the growth and development of plants [26]. Cutierrez found that volatile organic compounds produced by *Bacillus* could promote root elongation and change the root structure of *Arabidopsis thaliana* [27]. In addition, the use of microorganisms to detoxify aflatoxins is a promising new technology with broad application prospects [2]. For example, the excellent adsorption capacity and natural fermentation function make the use of *Lactobacillaceae* and *Saccharomyces* essential in the process of detoxifying food. *Lactobacillus rhamnosus strain* is an excellent biosorption species. The combination of heat-treatment and anaerobic solid fermentation can remove 100% of AFB1 [28]. Research proved the capacity of *L.acidophilus* to bind AB1 and AM1 in cow’s milk [29], which can reduce AFM1 (aflatoxin M1) and potentially decrease toxins in yogurt to a safe concentration for consumption (below 0.05 µg/kg) [30].

*Bacillus velezensis* is a plant growth-promoting bacterium that can inhibit plant pathogens [31], but can also produce a variety of antibacterial proteins and lipopeptide antibacterial substances with broad-spectrum antibacterial activities. Li et al. demonstrated that lipopolysaccharide produced by *B. velezensis* can not only inhibit spore of *A. flavus* germination and cause abnormal mycelium expansion and cell rupture, but can also significantly down-regulate the genes (*aflK*, *aflR*, *Vea*, and *omtA*) of aflatoxin synthesis pathways [32]. KONGQ et al. carried out in vivo and in vitro experiments on a marine *B. velezensis* strain isolated from the Yellow Sea in eastern China to test its effect on post-harvest corruption of peanuts caused by *A. flavus*. The results showed that the *B. velezensis* had significant biological control effects in vivo, and could reduce the biosynthesis of aflatoxin by inhibiting the transcription of the *aflR* gene and *aflS* gene [33]. *B. velezensis* DY3108, isolated from the soil, could detoxify fungal contamination in food by degrading 90% of aflatoxins B1 [34]. Vahidinasab et al. [35] reported that the *B. velezensis* UTB96 strain showed the highest antifungal activity against *A. flavus* and was capable of degrading aflatoxin. *B. velezensis* as a probiotic can compete with aflatoxigenic strains, either for nutrients and space, or for the degradation of aflatoxins by enzyme production [36].

Thus far, the molecular mechanism of *Bacillus* against *A. flavus* is not clear. The aim of this study is to find potential biocontrol agents against *A. flavus* growth and aflatoxin synthesis and understand the antagonistic mechanism. We found two bacterial strains, B1 and B2, isolated from the rhizosphere soil of *camellia sinensis* had significant antagonistic activities against *A. flavus* and identified them as the *Bacillus tequilensis* strain B1 and *Bacillus velezensis* strain B2. Transcriptome analysis showed that some genes related to *A. flavus* growth and aflatoxin synthesis were differentially expressed and 16 genes in the aflatoxin synthesis gene cluster showed down-regulation trends when inhibited by *Bacillus velezensis* strain B2. This provided the foundation for the more effective biological control of *A. flavus* and aflatoxin synthesis.

## 2. Materials and Methods

### 2.1. Fungal Strain

The fungal strain *Aspergillus flavus* was obtained from Prof. Xiufang Hu at the microbiology lab of Zhejiang Sci-Tech University. *A. flavus* was cultured on a PDA (Potato Dextrose Agar) medium and incubated at 28 °C [32,37,38] for 10 days.

### 2.2. Isolation of Bacterial Isolates from Camellia sinensis

The rhizosphere soil of *Camellia sinensis* was collected from tea garden in Meijiawu, Hangzhou, Zhejiang province. Samples were collected aseptically into Ziplock bags and marked labels, respectively, which were then transported to the laboratory for analysis. We weighed 10 g of rhizosphere soil and added a conical flask containing 90 mL sterile water to make a 10^−1^ soil suspension, and then we made 10^−2^~10^−9^ soil suspensions by serial dilution. 100 μL of 10^−7^, 10^−8^, and 10^−9^ soil suspensions were, respectively, spread on the LB (Luria Bertani) medium and then incubated at 37 °C for 4 days. Finally, 10 bacterial strains were obtained by isolation and purification.

### 2.3. Screen for Potential Biocontrol Agents against A. flavus 

Using a sterile cork-bore, mycelial disc (7 mm in diameter) of *A. flavus* from a 10-days-old culture on a PDA plate was placed in the center of a freshly prepared PDA plate (9 cm in diameter). Additionally, bacterial isolates were inoculated in an LB liquid culture and incubated at 37 °C for 16 h with shaking at 200 rpm, and 20 μL of the bacterial suspensions were, respectively, added to the three aseptic filter paper pieces at a distance of 3 cm from the mycelial discs. Control tests were carried out by only inoculating *A. flavus*. All the plates were placed in a 28 °C incubator, and the diameters of *A. flavus* were measured everyday from 2 d to 7 d. The relative inhibition rates were calculated as (C-T)/C × 100% [39], where C denotes the radius of *A. flavus* in the control and T denotes the radius of *A. flavus* in the treatment. All the experiments were done in triplicate.

### 2.4. Identification of Bacterial Isolates That Show Antagonism against A. flavus

The antagonistic bacterial isolates were identified both morphologically and genetically. For morphology, the isolates were cultured on plates at 37 °C for 4 days for colony morphology observation, and the pure colonies were stained with gram and observed using light microscopes. At the same time, the pure cultures were used to extract genomic DNA using Wizard Genomic DNA Purification Kit (Solario) according to the manufacturer’s instructions. 16SrDNA sequences of bacterial isolates were amplified by using the forward primer (TCCGTAGGTGAACCTGCGG) and reverse primer (TCCTCCGCTTATTGATATGC) and sequenced by the Sangon Biotech Company (Shanghai, China). Phylogenetic trees were constructed using the neighbor-joining method (MEGA 7) based on 16SrDNA gene sequences. Bootstrap resampling analysis was performed to estimate the confidence of the tree topologies.

### 2.5. Conidiation Assays of A. flavus Antagonized by B1 and B2

Conidial yields were determined, as previously described [40]. Briefly, 10 μL of the conidial suspension (1 × 10^7^ conidia mL^−1^) of *A. flavus* were inoculated on the center of PDA plates (90 mm in diameter). Additionally, three asptic filter paper pieces equidistantly spaced around 2.5 cm with *A. flavus* were then inoculated with 10 μL bacterial suspension of B1 and B2, which were cultured in LB at 37 °C for 16 h (200 rpm), while treatments inoculated with 10 μL of sterile LB were used as negative controls. All the PDA plates were placed in a 28 °C incubator. After 7 days culture, three agar discs (5 mm) of *A. flavus* were cut using a sterile cork-bore and respectively put into 1 mL 0.01%TitonX-100 (*v*/*v*). Conidial yields of *A. flavus* were determined after ultrasonication (30 Hz for 10 min) of the agar discs (JXFSTPRP-24 Shanghai Jingxin, Shanghai, China).

### 2.6. Transcriptomic Analysis of A. flavus Inhibited by B2 Using RNA-Seq

RNA-seq was used to profile transcriptomes of *A. flavus* inhibited by the *B. velezensis* strain B2. Briefly, B2 was inoculated in LB liquid culture and incubated at 37 °C for 16 h with shaking 200 rpm, and 20 μL of the bacterial suspensions were inoculated in the center of a freshly prepared PDA plate (9 cm in diameter), mycelial discs (7 mm in diameter) of *A. flavus* from a 10-days-old culture on a PDA plate and were respectively placed at a distance of 3 cm from the B2. Treatment of *A. flavus* without B2 inhibition was used as a negative control. After 4 days incubation at 28 °C, the fungal biomass of *A. flavus* were, respectively, subjected to RNA extraction with TRIzol reagent (Life Technologies, Carlsbad, CA, USA).

Construction of cDNA libraries and sequencing with the Illumina HiSeq 2000 platform were performed by Frasergen (Shanghai, China). After paired-end sequencing, clean reads were used for gene expression quantification using the RPKM method. Differential expression analysis was performed with edgeR v 3.24 software using the cut-offs of the adjusted *p*-value 0.05 and a greater than or equal to twofold change [41].

## 3. Results

### 3.1. Screen for Potential Biocontrol Agents against A. flavus

To screen for potential biocontrol agents, we isolated 10 bacterial strains from the oilseed plant of *Camellia sinensis,* which is very susceptible to be infected and contaminated by *A. flavus* and its production of aflatoxins (Appendix A). All strains (B1~B10) were tested to determine if they could inhibit the growth of *A. flavus* using the plate confrontation antagonism test. Only strains of B1 and B2 showed significant inhibiting effects on *A. flavus* (Figure 1A,B), and their relative inhibition rates were 34.22% and 35.72%, respectively (Appendix A).

### 3.2. Identification of Strains of B1 and B2

The resulting colony of B1 was 2.0–3.0 mm in diameter, smooth, circular, flat and white after being grown on LB media at 37 °C for four days, and B2 was 2.0–3.0 mm in diameter, smooth, circular but with an irregular edge, convex and white. B1 and B2 were both Gram-positive (Figure 2A). BLASTn searches based on the 16SrDNA gene sequences of B1 and B2 appeared that they matched *Bacillus tequilensis* and *Bacillus velezensis* with 99% similarity, respectively. Phylogenetic trees constructed by the neighbor-joining method showed that B1 and B2, respectively, clustered with the *Bacillus tequilensis* strain 10b (NR 118290.1) and *Bacillus velezensis* strain CBMB205 (NR 116240.1), indicating that B1 and B2 are likely isolates belonging to *Bacillus* (Figure 2B,C).

### 3.3. Conidiation of A. flavus Antagonized by B1 and B2

In *A. flavus*, conidiation is accompanied by aflatoxin synthesis [32,42]. Compared to the control, conidial yields of *A. flavus* antagonized by B1 and B2 significantly decreased (*p* < 0.05) (Figure 3). Therefore, it was speculated that B1 and B2 may secrete some compounds that inhibit the conidiation of *A. flavus*, and then reduce the synthesis of aflatoxin.

### 3.4. Overview of RNA-Seq Analysis and DEGs Identification of A. flavus Antagonized by B2

To find the genes that may be related to the synthesize of aflatoxins and regulated by bacterial isolate B2, the transcriptomes of *A. flavus,* which was inhibited by B2, were profiled by Illumina Hiseq 2000 RNA-Seq (*A. flavus* without antagonism as a negative control), three biological replicates and eight datasets were established. In this section, we compared a set of differentially expressed genes (DEGs) between *A. flavus* antagonized by B2 and *A. flavus* alone. Additionally, ‘up-regulated genes’ are genes with higher expression levels when *A. flavus* antagonized by B2, ‘down-regulated genes’, are those with lower expression levels under the same conditions. When *A. flavus* was antagonized by B2, there were a total of 185 DEGs, including 151 up-regulated and 34 down-regulated genes (Figure 4).

### 3.5. GO Terms Analysis of DEGs

Among the 185 genes that were differentially expressed in *A. flavus* inhibited by B2, 30 GO (Gene Ontology) terms were enriched, including ten in biological process, ten in cellular component and ten in molecular function (Figure 5). A GO term for aflatoxin biosynthesis was enriched for two up-regulated genes of aldo-keto-reductase family and cytochrome P450 related genes (*AFLA_3422* and *AFLA_860*) (Table 1). Aldehyde-ketoreductase family genes synthesize NAD(P)-linked oxidoreductases that catalyze the reduction of NAD(P)H to NAD(P)^+^. Cytochrome P450 gene synthesizes cytochrome P450 monooxygenase. In the synthesis of aflatoxin, the role of this enzyme is to catalyze the conversion of O-methyl variegated toxin and dihydro-O-methyl variegated toxin into aflatoxins B1, G1, B2, and G2, but the expression of their activities requires NADPH as a cofactor, and the level of intracellular NADPH/NADP^+^ affects the expression of aflatoxin biosynthesis regulator gene *aflR* [11]. Therefore, it is speculated that the compound produced by B2 can promote the production of NAD(P)-linked oxidoreductase and catalyze the reduction of intracellular NAD(P)H to NAD(P)^+^, thereby inhibiting the production of aflatoxin.

### 3.6. KEGG Enrichment Analysis of DEGs

The results of the KEGG enrichment of 185 DEGs showed that they were involved in 45 metabolic pathways. Among them, genes in the biosynthetic pathway of amino acids and the metabolic pathway of carbohydrates were mainly up-regulated, and they were consistent with the results of the research reported previously [4]. Ribosomes are the cellular factories responsible for making proteins. Additionally, according to the results of transcriptome sequencing, genes related to ribosome synthesis were mainly up-regulated in *A. flavus* (Table 1). Therefore, it was speculated that the active components secreted by B2 may lead to protein damage and membrane structure destruction in *A. flavus*, and *A. flavus* synthesizes proteins from other pathways to maintain its own basic growth. The pathways enriched for down-regulated genes mainly include the biosynthetic process, energy metabolism, transcription, folding, transport, and catabolism (Table 1); these genes were down-regulated may affect ribosome assembly and the protein synthesis of *A. flavus*, thus resulting in growth that was inhibited and the biosynthesis of aflatoxins was reduced.

## 4. Discussion

*Aspergillus flavus* is an universal pathogen of crops and it produces aflatoxin in the seeds of a variety of crops before and after harvest, which poisons humans and animals through the food chains. Currently, many strategies have been commonly used to control aflatoxin production. Preharvest control has relied on identifying resistant crop lines, planting regionally adapted cultivars and planting at appropriate seed densities, thus limiting insect damage and so on. However, these measures are not always efficient to avoid aflatoxin formation [10,43]. More and more biocontrols have been developed. As one strategy, aflatoxin non-producing strains of *A. flavus* excluded competitive aflatoxin-producing *Aspergillus* species to decrease crop aflatoxin contamination [9]. Further, aflatoxin biocontrol by beneficial microorganisms, including bacteria, actinomycetes and so on, is considered to be one of the most promising practices [44].

*Camellia* is a kind of oil plant and seed vulnerable to *A. flavus* infection and aflatoxin contamination. To screen for potential biocontrol agents that inhibit *A. flavus* growth and reduce aflatoxin biosynthesis, we isolated 10 bacterial strains from the leaves, flowers, seeds and rhizosphere soil of *camellia*. We found that only bacterial strains of B1 and B2 can inhibit the growth of *A. flavus* among the 10 bacterial strains. Some studies showed that volatile organic compounds, such as 2, 3-butanediol produced by soil-borne endophytic bacteria, increase plant pathogen resistance and promote plant growth [45,46]. However, B1 and B2 did not inhibit *A. flavus* by producing volatile compounds (Appendix A), and they may have other antagonistic mechanisms.

BLASTn searches and phylogentic trees based on the 16SrDNA gene sequences showed that the bacterial strains B1 and B2 clustered well with the *Bacillus tequilensis* and *Bacillus velezensis*, respectively. In addition, B1 and B2 had typical morphological characteristics that were similar to those of *Bacillus*. Therefore, B1 and B2 are both identified as the *Bacillus* genus. Members of the *Bacillus* genus produce abundant biologically active molecules, such as lipopeptides (LPs), polyketides, peptides, phosphatides, polyenes and so on, and they play important roles in controlling plant pathogens [47]. In a previous study, an antagonistic strain of *Bacillus velezensis,* with obvious anti-*Aspergillus flavus* fungi activity, was isolated from the surface of healthy rice, and the main components of LPs produced by this strain were identified by HPLC-MS analysis as fengycin and iturins, and they had shown that LPs can inhibit spore germination and even cause abnormal hyphal expansion and cell rupture [44]. Therefore, we guessed that B1 and B2 may produce similar antimicrobial compounds to inhibit *A. flavus* growth.

The study found that the level of intracellular NADPH/NADP^+^ has a certain effect on the expression of *aflR*, and NADPH is a cofactor for the expression of cytochrome P450 monooxygenase activity. In *A. flavus*, cytochrome P450 monooxidase catalyzes the conversion of O-methyl versicolor and dihydro-O-methyl versicolor to aflatoxin B1, G1, B2, and G2, respectively [11]. We found that there were 185 DEGs, including 152 up-regulated genes and 34 down-regulated genes in *A flavus* antagonized by the *Bacillus velezensis* strain by the transcriptome analysis. GO term and KEGG enrichment analysis were performed on these DEGs, and we found that the GO term of aflatoxin biosynthesis was enriched for two up-regulated genes. They were namely aldosterone reductase family genes and cytochrome P450 related genes, which synthesize NAD(P)-linked oxidoreductase and cytochrome P450 monooxygenase, respectively. NAD(P)-linked oxidoreductase catalyzes the reduction of NAD(P)H to NAD(P)^+^. Therefore, it is speculated that the compound produced by B2 can promote the production of NAD(P)-linked oxidoreductase and catalyze the reduction of intracellular NAD(P)H to NAD(P)^+^, thus inhibiting the monooxygenase activity of cytochrome P450 and even the aflatoxin production. The KEGG enrichment analysis found that related genes in amino acid biosynthesis pathway, carbohydrate metabolism pathway and ribosome synthesis pathway were all up-regulated, and genes in the energy metabolism pathway, transcription, folding, transport pathway, biosynthesis and catabolism pathway were down-regulated. It was speculated that the active components secreted by B2 lead to protein damage and membrane structure disruption in *A. flavus* and *A. flavus* synthesizes proteins through other pathways to maintain its own basic growth. However, the down-regulated genes may affect ribosome assembly and protein synthesis, thus resulting in growth that was inhibited and biosynthesis of aflatoxins that was reduced.

Aflatoxins are PKS-derived mycotoxins synthesized from a large cluster, which is composed of approximately 30 different genes. The expression level of each gene in this pathway directly affects the amount of aflatoxins synthesis. A*flA*, *aflB* and *aflC* are required to synthesize norsolorinic acid, which is the first stable aflatoxin precursor [10]. VeA, as the core protein of the velvet complex, regulates the growth and development of *A. flavus,* including participating in conidiogenesis and the publication of sclerotia [48]. A*flR* and *aflS* are the regulatory genes of aflatoxin biosynthesis, which play a key role in the regulation of aflatoxin biosynthesis [49,50]. A*flK* Paticipates in the conversion of VAL to VERB, which is a key step in the formation of aflatoxin and blocks the bifuran ring of aflatoxin [51]. Af*lP* (*omtA*) gene encoding protein is a key enzyme in the late stage of aflatoxin synthesis [52]. Transcriptome analysis of *A. flavus,* when inhibited with B2, indicated that 16 genes in the gene cluster of aflatoxin synthesis show down-regulation trends (Appendix A). This further indicates that the secondary metabolites produced by B2 can down-regulate the transcription of related genes and reduce the production of aflatoxin.

At present, non-toxigenic-strain of *A. flavus* or non-toxingenic-strain of *Parasitic Aspergillus* are often applied to the soil to competitively inhibit the growth of aflatoxin-producing strains, so as to effectively prevent aflatoxin pollution before and after harvest [53]. Jia chang Li uses starch, sodium alginate and thermosensitive gel as a matrix, and non-toxigenic-strain of *A. flavus* suspension as the source of biocontrol bacteria and prepares temperature-responsive immobilized microspheres by immobilized microtechnology, which can continuously release biocontrol bacteria at a suitable temperature in the field and effectively prevent aflatoxin contamination [54]. However, in practical application, the requirement of non-toxigenic-strain of *A. flavus* is very high. It can not only produce toxin but must also have strong vitality and competitiveness. Such strains are not easy to obtain in the environment [55]. *Bacillus* is a kind of aerobic or facultative anaerobic bacteria. It is not only an important microbial population in soil and plant surface rhizosphere, but also a common endophytic bacterium in plants. It has the characteristics of a wide distribution and is easy to obtain [56]. Secondary metabolites produced by *Bacillus* in plant rhizosphere can not only inhibit the growth and toxin production of *A. flavus*, but can also promote plant growth [25]. Previous studies have reported the usefulness of *B. velezensis* as a probiotic in the aquaculture industry [57,58,59]. Due to its wide distribution in nature and enrichment in its own metabolites [60], *B. velezensis* may have the potential to be used as a probiotic in animal feed. Compared to non-toxigenic-strains of *A. flavus*, *Bacillus* strains are easier to culture and obtain and have stronger environmental adaptability. We can prepare liquid bacterial agent B2 and add a large amount of it to rhizosphere soil of tea to inhibit *A. flavus*, which is a saprophytic soil fungus, and has a better inhibitory effect on the actual crop groups in the field. *Bacillus* can also produce a variety of antibacterial proteins and lipopeptide antibacterial substances, which have broad-spectrum antibacterial activity [28]. The lipopeptide produced by *B. velezensis* B2 can be isolated and purified by the fermentation process and made into a preparation to inhibit the growth of *A. flavus* and aflatoxin production during grain storage [56]. To sum up, *B. velezensis* B2 isolated from *camellia* has great potential to become a biocontrol bacterium for the prevention of aflatoxin pollution.

## Figures and Tables

**Figure 1 foods-11-03620-f001:**
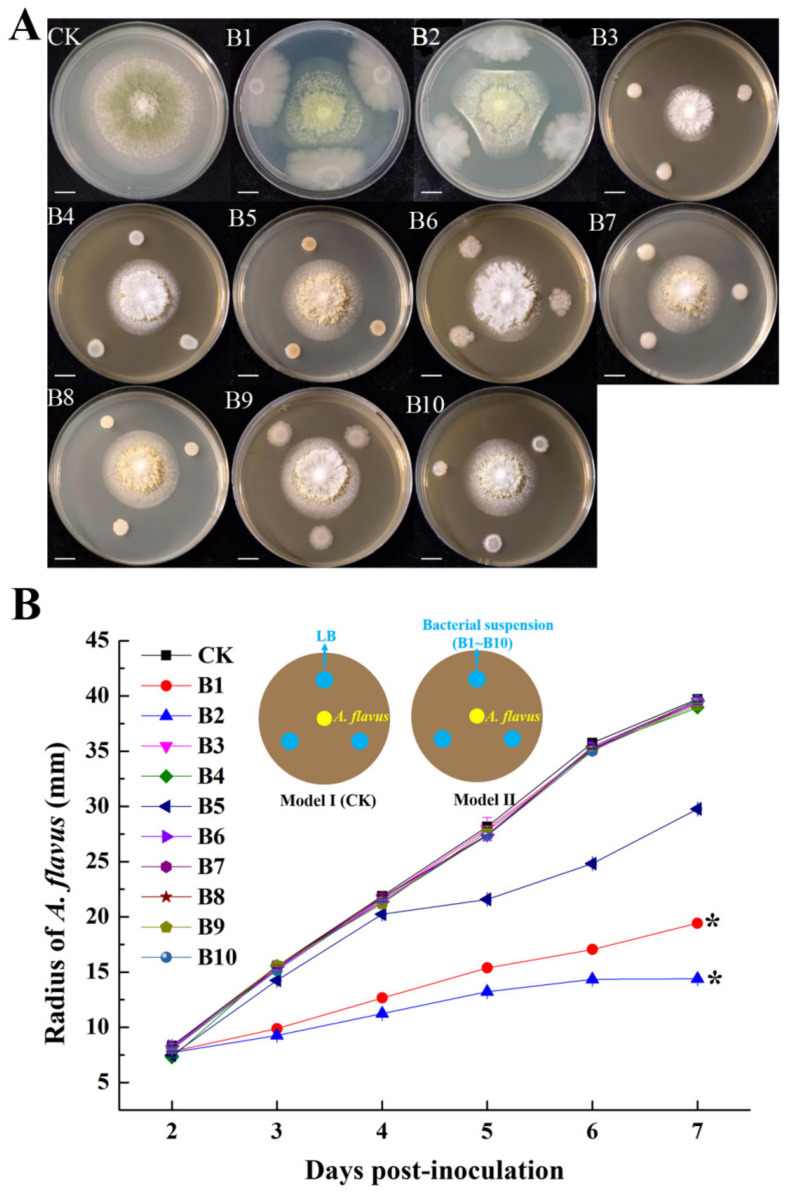
Inhibitory effect of bacterial isolates against *A. flavus*. (**A**) Plate confrontation antagonism test and colony pictures were taken at four days post-inoculation. Scale bars represent 10 mm. (**B**) Growth of *A. flavus* when *A. flavus* only (CK) and inhibited by different bacterial isolates such as B1, B2, B3, B4, B5, B6, B7, B8, B9 and B10. The star (*) indicates that the growth of *A. flavus* when inhibited by the bacterial isolate is significantly slower than CK (*p* < 0.05). Inhibitory effect assays were repeated three times with three replicates per repeat.

**Figure 2 foods-11-03620-f002:**
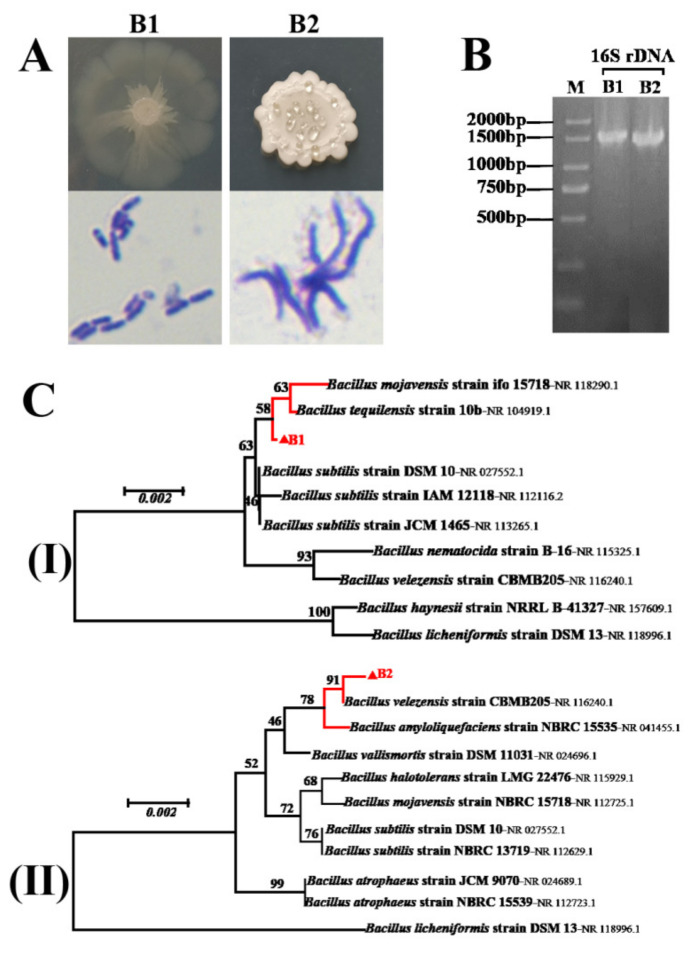
Identification of bacterial isolates B1 and B2 from morphologies and genetics. (**A**) Morphology and Gram staining of B1 and B2. Colony pictures were taken at four days post-inoculation by applying 5 μL of a bacterial suspension on the center of a LB plate (diameter 9 cm). (**C**) Phylogenetic trees were constructed by the neighbor-joining method based on 16S rDNA gene sequences (**B**), respectively showing the phylogenetic relationships of strain B1 (I) and B2 (II) and related species; numbers on branches are the bootstrap values that were based on 1050 replicates, and the scale bar indicates 0.002 substitutions per nucleotide position.

**Figure 3 foods-11-03620-f003:**
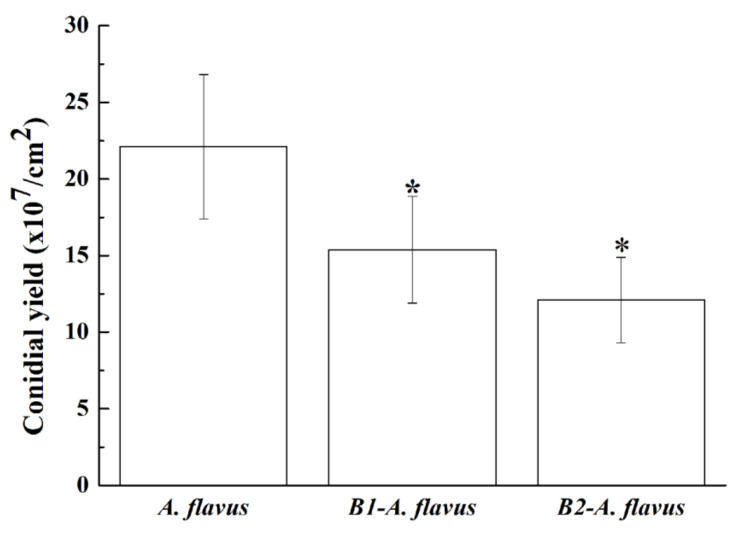
Conidial yields (mean ± SE) of *A. flavus* and *A. flavus* antagonized by B1 and B2. Conidial yields were determined at 10 days post-inoculation. The star (*) indicates that *A. flavus* inhibited by B1 and B2 both produce significantly less conidia than the control (*p* < 0.05).

**Figure 4 foods-11-03620-f004:**
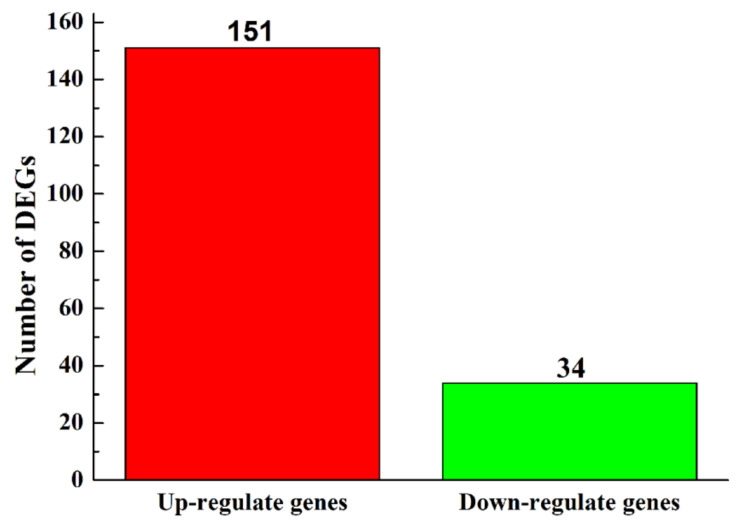
DEGs analysis of *A. flavus* antagonized by B2 compared to no antagonism.

**Figure 5 foods-11-03620-f005:**
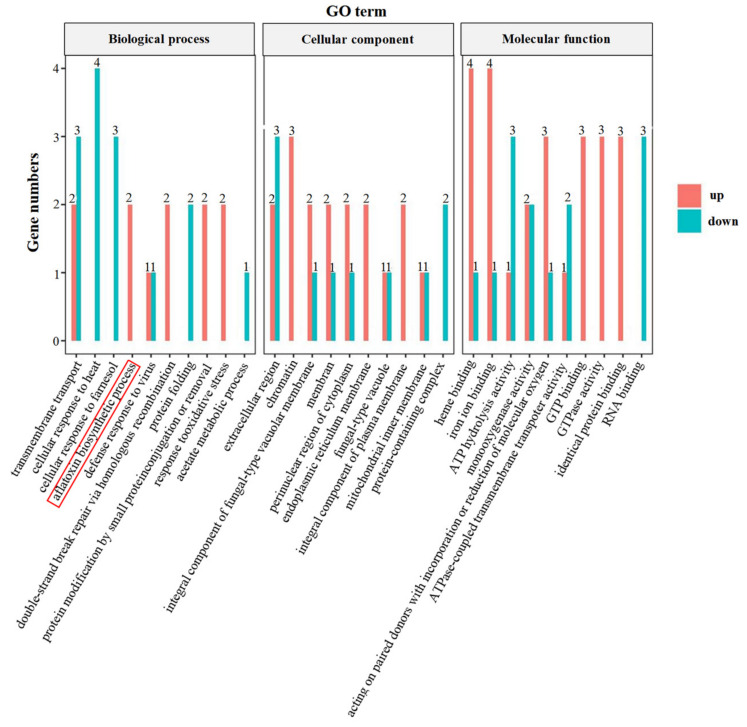
GO terms analysis of DEGs in *A. flavus* antagonized by B2. Red columns stand for up-regulated genes and blue for down-regulated genes. The GO term enriched for core DEGs was marked with a red box.

**Table 1 foods-11-03620-t001:** Related genes involved in the growth of *A. flavus* and aflatoxin synthesis.

Pathway	Gene ID	Gene Description	Log2FC	Style
Aflatoxin biosynthetic process	*AFLA_3422*	Aldo/keto reductase family	9.50	up
	*AFLA_860*	Cytochrome P450	8.23	up
Ribosome synthesis	*AFLA_23553*	50s ribosome-binding GTPase	11.03	up
	*AFLA_19265*	50s ribosome-binding GTPase	11.05	up
	*AFLA_7027*	50s ribosome-binding GTPase	9.39	up
Biosynthetic process	*AFLA_1792*	Biosynthesis of secondary metabolites	−2.56	down
	*AFLA_2285*	Biosynthesis of secondary metabolites	−2.24	down
Energy metabolism	*AFLA_2285*	Glycolysis/gluconeogenesis	−2.24	down
Folding, sorting and degradation	*AFLA_18974*	Protein processing in Endoplasmic reticulum	−2.17	down
	*AFLA_12766*	Protein processing in Endoplasmic reticulum	−1.81	down
Catabolism	*AFLA_3999*	Carbon metabolism	−1.65	down

## Data Availability

Not applicable.

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
