# Peer review of "Differential Expression of Genes Related to Growth and Aflatoxin Synthesis in Aspergillus flavus When Inhibited by Bacillus velezensis Strain B2"

_foods, 2022, doi:10.3390/foods11223620_

Round 1

Reviewer 1 Report

The present manuscript entitled “Inhibition of growth and aflatoxin synthesis of Aspergillus flavus by Bacillus velezensis strain B2” present data of in the possible inhibition of growth and aflatoxin synthesis of Aspergillus flavus by the antagonist bacteria Bacillus velezensis strain B2 on the possible impacts of Bacillus velezensis strain B2 on both the growth of Aspergillus flavus and aflatoxin contamination. The data is interesting and show that Bacillus velezensis strain B2 might have some inhibition effects on both the growth of Aspergillus flavus and the synthesis of aflatoxins by A. flavus. The experiment seems to be carried out well and the data looks sound to be published in Foods. However, there are some issues that need to be corrected before publication. Perhaps starting with the title, which should indicate that Bacillus velezensis strain B2 might have some inhibition effects on growth and aflatoxin synthesis in Aspergillus.

I consider that the data on gene expression is good and appropriate. However, the experiments to determine the gene expression were incubated at 26 C, which might be in the lower range of both optimal growth and aflatoxin synthesis (28 – 30 C). Now the question is, would incubating at the optimal temperature of 30C change the gene expression profile? Would B2 inhibits A. flavus growth at the same levels when incubated at optimal temperatures?

My main concern with this manuscript is that the supposed inhibition effects of Bacillus velezensis strain B2 are all speculation based on other possible effects of gene expression. The authors do not show data indicating direct evidence of such inhibitions. The manuscript will be stronger if the relationship of the up and down regulated genes with aflatoxin synthesis is presented. Furthermore, the data show a quite low inhibition of A. flavus growth, which I don’t think will be an effective biocontrol agent in real crop environments.

I wonder, and the authors should discuss, how this could be implemented out of the laboratory in real field crops sets. Also need to discuss if this biocontrol agent will have better control response than the competitive displacement of toxigenic strains using non-toxigenic strains of A. flavus, which is being use commercially. Also need to discuss comparing these two technologies. The authors put much emphasis discussing other aflatoxin control methods, including fungicide use, which are not really used commercially, and fail to discuss the only method of control that has been proven to be effective commercially.

Other concerns.

Paragraph in the introduction section regarding methods of control (L44-L51) are based on fungi other than aflatoxin producing Aspergillus and misses the point. Fungicide use is not an option to control Aspergillus. And they are not use for that purpose. The citation for that purpose is not adequate. Should cite a different work discussing the methods of control used for aflatoxin producing fungi. The authors fail to address the only control method, either biological or otherwise, that has been used commercially. They should discuss what methods of aflatoxin control are actually being used commercially and not speculate with methods that have some promises in the lab but has not been proven successful (L52-L69). They should discuss how this bacterial inhibition method could compare to the commercially available methods of control as the competitive displacement using non-aflatoxigenic strains of Aspergillus flavus.

Also, the paragraph in the discussion section L291-L301 is not relevant to the discussion, it looks more like a review. Perhaps should be better to move it to the introduction section or removed.

Reviewer 2 Report

This paper describes the inhibitory effects of Bacillus spp. on Aspergillus flavus. Some experiments seem to be interesting, but the explanations are not clear and hard to understand for readers. For example, in Figure 1, I wonder why you explain that only strains of B1 and B2 showed inhibitory effects on the growth Aspergillus flavus from this experiment.
<Specific comments>
Abstract: Aspergillus flavus is generally saprophytic. I recommend to rewrite.

Reviewer 3 Report

Line 18 and 19 - please continue the statement ...." by mechanisms as...".

Line 31, 142: space is needed

Line 37: aflatoxins-contiminated "contaminated"

Line 77: Micromeria graeca should be in italic

Please state in the introduction the aim of the study. 

Please use the italic style for Camellia sinensis anywhere it appears in the text.

Line 102: it is not clear how the dilutions were made. From what? In what solvent?

Line 117 - why is the equation not numbered?

Line 134: "previously"

Line 135: isn't 107?

Line 220, 242: the first time an abbreviation appears in the text, it should be explained

Please state how Kyoto Encyclopedia of Genes and Genomes was used. 

In the Results part: is about  "Camellia sinensis" please state that.

Please also read the text carefully, it is full of English and punctuation typos. 

There are a lot of relevant and recent references that are not discussed. 

Round 2

Reviewer 2 Report

Description about A. flavus becomes improved. Fig. 1 explanation becomes fair.

Author Response

I am very grateful for your comment and suggestion.

Reviewer 3 Report

Dear authors,

The paper is much improved but still needs some final improvements prior to publication. 

Please reconsider the paper for English spelling! (i.e. a saprophytic soul fungus; the 16SrDNA gene etc).

The reference list must be improved. There are some relevant and new papers that were not cited. For example: https://doi.org/10.1007/s12275-021-1161-1, https://doi.org/10.3390/jof8020190, https://doi.org/10.3390/toxins13010046, https://doi.org/10.3390/microorganisms10071278.

After conducting these changes I belive that the paper is suitable for publication.
